

# Comparative anatomical and transcriptomic analyses of the color variation of leaves in *Aquilaria sinensis*

Jiaqi Gao[1,2], Tong Chen[1], Chao Jiang[1], Tielin Wang[1], Ou Huang[3], Xiang Zhang[1] and Juan Liu[1]

[1] National Resource Center for Chinese Materia Medica, China Academy of Chinese Medical Sciences, Beijing, China
[2] School of Pharmacy, Jiangsu University, Zhenjiang, China
[3] Guangdong Shangzhengtang Group Co., Ltd, Dongguan, China

## ABSTRACT

Color variation in plant tissues is a common phenomenon accompanied with a series of biological changes. In this study, a special-phenotype *Aquilaria sinensis* (GS) with color variation of leaf was firstly reported, and DNA barcode sequences showed GS samples could not be discriminated clearly with the normal *A. sinensis* sample (NS), which suggested that the variety was not the cause of the GS formation. To reveal the characteristics of GS compared to NS, the anatomical and transcriptome sequencing studies were carried out. In microscopic observation, the leaves of golden-vein-leaf sample (LGS) and normal-vein-leaf sample (LNS) showed significant differences including the area of the included phloem in midrib and the thickness parameters of palisade and spongy tissues; the stems of golden-vein-leaf sample (SGS) and normal-vein-leaf sample (SNS) were also different in many aspects such as the area of vessels and included phloem. In addition, the structure of chloroplast was more complete in the midrib of LNS than that of LGS, and some particles suspected as virus were found through transmission electron microscope as well. Genes upregulated in LGS in contrast with LNS were mainly enriched in photosynthesis. As for stems, most of the genes upregulated in SGS compared to SNS were involved in translation and metabolism processes. The pathways about photosynthesis and chlorophyll metabolism as well as some important transcription factors may explain the molecular mechanism of the unique phenotypes of leaves and the genes related to suberin biosynthesis may result in the difference of stems. In addition, the genes about defense response especially biotic stress associated with numerous pathogenesis-related (PR) genes upregulated in LGS compared to LNS indicated that the pathogen may be the internal factor. Taken together, our results reveal the macro- and micro-phenotype variations as well as gene expression profiles between GS and NS, which could provide valuable clues for elucidating the mechanism of the color variation of *Aquilaria*.

Corresponding authors
Tong Chen, chent@nrc.ac.cn, chentong_biology@163.com
Juan Liu, juanliu126@126.com

## INTRODUCTION

Agarwood is a resinous, fragrant wood, which is highly valued for its use in medicine, perfumes, and incense across Asia, Middle East and Europe (*Takemoto et al., 2008*). It is produced by species of tropical trees of the genus *Aquilaria*, whose population is dramatically declining due to overexploitation. Furthermore, all of the *Aquilaria* species, including *A. sinensis* (Lour.) Gilg which is the only certified source for agarwood listed in China Pharmacopoeia (*Committee, 2020*), are conserved under the Convention on International Trade in Endangered Species of Wild Fauna and Flora (CITES, http://www.cites.org). However, the source of wild agarwood is facing serious depletion drain owing to its slow and infrequent formation and uncontrolled collection in forests. Fortunately, *A. sinensis* has been widely cultivated in the Dongguan area of China, as early as the Tang and Song Dynasties. During long-term cultivation, some phenotypic differences of *A. sinensis* have been showed up, some of which could be seemed as potential varieties worth to be explored and bred. Fortunately, we firstly investigated the color variation of leaves existed in *A. sinensis*, which is also the first report about color variation in genus *Aquilaria*. However, there is no further information about this phenotype and potential value in agarwood industry.

In the plant kingdom, color variation has been widely reported in leaf, culm, flower, fruit, seed coat and other tissues derived from different plants (*Ahmed, Hayashi & Yazawa, 2004*; *Barazani et al., 2019*; *Brand et al., 2014*; *Xia et al., 2015*; *Yang et al., 2010*). Some phenotypes about color variation could be found in the whole tissue while others could show diversified phenotypes such as color stripes or banding mosaic (*Wang et al., 2019*; *Xia et al., 2015*). These phenotypes emerge with a series of changes in colored substances such as chlorophyll, carotenoid, anthocyanins, flavonoids or other pigments (*Cheng et al., 2018*; *He et al., 2011*; *Khandagale & Gawande, 2019*; *Xia et al., 2015*). Accompanied with the changes in appearance, a battery of characteristics in botany or ecology could be different. For example, the flowers of *Eruca sativa* with different color display diverse attraction for their main pollinator because of the discrimination of the scents in flowers of different colors (*Barazani et al., 2019*). In *Capsicum annuum* L., the plant samples differing in leaf color are accompanied with different combinations of pigments which could influence the resistance of these plants against whitefly (*Cheng et al., 2018*) which is a kind of pest damaging plants seriously (*Ullah & Lim, 2016*). In the studies about the color variation in fruit like tomato, the skin pigmentation, carotenoid biosynthesis and ripening-associated chlorophyll degradation were claimed as main factors in the formation mechanism of fruit color in tomato (*Chattopadhyay et al., 2021*). *SlMYB12*, *IDI1*, *PSY1*, *CrtISO*, *CYC-B*, *LCY-E*, and *SGR* were found to be important in the three pathways mentioned in color variation in tomato (*Chattopadhyay et al., 2021*). Regarding leaves, it was common that the color mutant would be used in studies about color variation (*Zhu et al., 2015*). Apart from the metabolism pathways like chlorophyll metabolism (*Zhu et al., 2015*), other factors including the recessive nuclear gene like *SiYGL1* (*Li et al., 2016a*; *Li et al., 2016b*), shock protein genes containing *HSP70*, *HSP90* (*Wu et al., 2018*) were also being studied.

Additionally, the plant with natural color variation is a kind of infrequent material which is worthwhile to be applied in ornamentation or scientific studies bringing considerable economic and practical value besides the influence and botanical functions to the plants themselves. For instance, the tea plants with green, yellowish, purplish leaves showed different active levels to the metabolism of catechins and the contents of theanine, caffeine and other chemicals with respect to the tea quality and characteristic suggesting the selection of breeding materials as new tea cultivars (*Li et al., 2016a*; *Li et al., 2016b*). Besides, some researchers revealed that a purple-colored rice landrace was a good resource for rice breeding because of its disease resistance, stress tolerance, enhanced nutritional values, etc. after the whole genome sequencing and comparative analysis of the unique colored rice (*Lachagari et al., 2019*). In addition, the plants with color variation of tissues are also important materials used to study the biosynthesis of some colored substances (*Gang et al., 2019b*). Thus, the *A. sinensis* plant with color variation should be a worthy and potential material to be investigated.

Herein, a unique-phenotype *A. sinensis* whose leaves showing a deeper green compared to the normal *A. sinensis* accompanied with golden (yellow) veins was firstly found, and DNA barcode sequences showed that the variety was not the cause of the GS formation. Additionally, the differences between the unique-phenotype *A. sinensis* and normal *A. sinensis* were analyzed using anatomical observation and transcriptome sequencing. We performed functional analysis of differentially expressed genes and assessed Gene Ontology (GO) annotations and Kyoto Encyclopedia of Genes and Genomes (KEGG) pathway enrichment analysis. Furthermore, the candidate genes related to photosynthesis and defense responses enriched from functional analysis were verified using quantitative real-time PCR. Overall, our integrated analysis allows for a better understanding of the molecular changes occurring in color variation in *A. sinensis*.

## MATERIALS & METHODS

### Plant materials

The golden-vein-leaf and normal-vein-leaf sample (GS and NS) *A. sinensis* plants were obtained from Dalingshan town, Guangdong province, China (latitude 22°45′43″N, longitude 113°48′45″E). Those plants grow in close proximity with similar environment and at a similar altitude of 70–80 m. The samples were all collected from six *A. sinensis* trees in which three trees were GS and the others were NS. Then, three leaves and three stems of GS as well as those of NS were collected from each three branches of trees and mixed respectively, with three biological replicates used for each group in DNA barcode sequencing and RNA sequencing. All of the samples were frozen in liquid nitrogen and stored at −80 °C for the next studies.

### DNA extraction and determination of chlorophyll content

The leaves were powdered in a mortar with a pestle in liquid nitrogen. The stems were sliced first, and then powered in liquid nitrogen according to the published methods (*Jiao et al., 2014*). Additionally, those samples were inserted into a 2 mL tube containing 1,000 μL buffer AP1, 8 μL RNase A, and 1% polyvinylpyrrolidone with incubation at 65 °C for 6 h
respectively. Cooling to room temperature, the 280 µL buffer P3 was added into the tubes and those samples were incubated for 2 h at −20 °C. Subsequent steps were conducted as the direction supplied by the manufacturer of EASYspin Plus Plant DNA Kit (Aidlab Biotechnologies Co., Ltd, Beijing, China). The determination of chlorophyll content was performed based on the method published before (*Sartory & Grobbelaar, 1984*).

### DNA barcode sequence amplification and phylogenetic tree analysis

ITS2 and *trn*L-*trn*F were used as barcode sequences to identify the species and the primers were adopted from a published paper (*Lee et al., 2016*). The PCR reaction system was also conducted as the previous report (*Lee et al., 2016*).

The ITS2 and *trn*L-*trn*F barcode sequences of GS and NS were aligned with *A. crassna, A. hirta, A. malaccensis, A. microcarpa, A. sinensis, A. yunnanensis, Gyrinops versteegii* and *Gonystylus bancanus* (Table S1) using Clustal W algorithm (*Larkin et al., 2007*). Subsequently, the phylogenetic tree was constructed based on neighbor-joining (NJ) method with 1,000 bootstrap replications. All the above analyses were operated with MEGA-X (10.1.8) (*Kumar et al., 2018*).

### Microscopic observation and measurement

The paraffin microsections were sliced by Leica CM1860 freezing microtome (Leica Microsystems Inc., Wetzlar, Germany) with Safranin O/Fast green staining subsequently. The microscopic observations of those microsections were conducted by ZEISS AX10 microscope (ZEISS corporation, Jena, Germany). ZEISS ZEN 2 lite software was applied to measure the parameters of the photomicrographs. The statistical differences were calculated by $T$-test using IBM SPSS Statistics 22. The ultrastructure was observed through transmission electron microscope H-7500 (Hitachi, Ltd., Japan).

### RNA extraction and transcriptome sequencing

The RNA was extracted from the *A. sinensis* samples using the EASYspin Plus Plant RNA Kit (Aidlab Biotechnologies Co., Ltd, Beijing, China) according to the manufacturer's instructions. The quality of extracted RNA was examined by NanoDrop 2000 spectrophotometer (Thermo Fisher Scientific, USA). Then, 0.5 µg RNA per sample was reverse-transcribed into single-stranded cDNA using cDNA Synthesis Kit (TaKaRa, Japan) as what the manufacturer's instructions suggested. Subsequently, the RNA template was eliminated and the second-strand cDNA synthesis was conducted with DNA polymerase, dNTPs, and RNase. After end repair, A-tailing and indexing ligation, the products were purified with PCR extraction kit, and amplified to be cDNA libraries. Finally, the sequencing was conducted on Illumina NovaSeq 6000. The data have been deposited in NGDC's Genome Sequence Archive (*Wang et al., 2017*), under accession number CRA002670 that is publicly accessible at https://bigd.big.ac.cn/gsa/browse/CRA002670. The de novo assembly was conducted by the Trinity method (*Grabherr et al., 2011*).

### Annotation and differentially expressed gene analysis

The annotations of transcriptome data were conducted by Diamond (*Buchfink, Xie & Huson, 2015*), KAAS (*Moriya et al., 2007*) and Blast2GO (*Gotz et al., 2008*). The gene

expression level was estimated with FPKM (*Trapnell et al., 2010*). Using DESeq2 (*Love, Huber & Anders, 2014*), when the parameters of genes expressed in different groups at $\log_2$(Fold Change) larger than 1 or less than −1, and false discovery rate (FDR) less than 0.05, they could be accepted as differentially expressed genes (DEGs). The GO and KEGG enrichments were conducted via R package clusterProfiler (*Yu et al., 2012*) with the custom data sets from the annotation results. For GO enrichment, 15,811 annotated genes were used as the background gene set. For KEGG, 4,865 annotated genes were used as the background gene set. In GO enrichment, the FDR screening threshold was set to be less than 0.05, while the value was set to be less than 0.1 in KEGG enrichment. The cluster analysis was conducted via the STEM clustering algorithm (*Ernst, Nau & Bar-Joseph, 2005*) and the cluster whose *P* value was less than 0.05 was considered as significant difference. All the plots were generated using one public on-line tool BIC (http://www.ehbio.com/Cloud_Platform/front/) or R scripts.

### Quantitative real-time PCR

The qRT-PCR was carried out with the TB Green$^@$ Fast qPCR Mix (TaKaRa, Japan) and LightCycler$^@$ 480 Real-Time PCR System (Roche, Switzerland). The reaction mixture contained 3 μL of nuclease-free water, 5 μL of TB Green Premix Ex Taq II (TaKaRa, Japan), 1 μL of cDNA, 0.5 μL of forward primer (10 μM), and 0.5 μL of reverse primer (10 μM). The program was set to run for 1 min at 94 °C, 40 cycles of 10 s at 94 °C, and 34 s at 60 °C. Melt curves were obtained by heating the samples from 60 °C to 95 °C at a rate of 1.0 °C/s (Table S9). The primers of genes were listed in Table S9 with a reference gene GAPDH (*Xu et al., 2016*). Each RNA sample was conducted with three biological repeats and three technical replicates. The relative gene expression levels were calculated through $2^{-\Delta\Delta CT}$ method (*Livak & Schmittgen, 2001*). The correlation analysis was conducted by Pearson correlation coefficient.

## RESULTS

### DNA barcode sequencing and phylogenetic tree analysis

The samples used in this study were all collected from Guanxiang Intangible Cultural Heritage Protection Park, while their phenotypes differed from each other. The golden-vein-leaf sample (GS) showed a color variation of leaf vein compared to the normal-vein-leaf sample (NS) and had deeper green and harder texture in leaves (Figs. 1A and 1B), and the contents of chlorophyll including chlorophyll a, chlorophyll b and carotenoid in the leaves of GS (LGS) were all higher than those in the leaves of NS (LNS) (Fig. 1C). Though the selected samples were cultured as the same species in the cultivation place, the phenotypes differences were so remarkable that the DNA molecular identification was employed to confirm the phylogeny relationships between GS and NS (Fig. S1). According to the previous report, the combination of ITS2+*trn*L-*trn*F was used as DNA barcode for *Aquilaria* species identification (*Lee et al., 2016*). The phylogenetic tree was constructed based on ITS2+*trn* L-*trn*F DNA barcode sequences in 8 closely related species, and one distantly related species *Gonystylus bancanus* as outgroup according to the previous study (*Lee et al., 2016*) (Fig. 1D). Most of the selected reference species had their separate

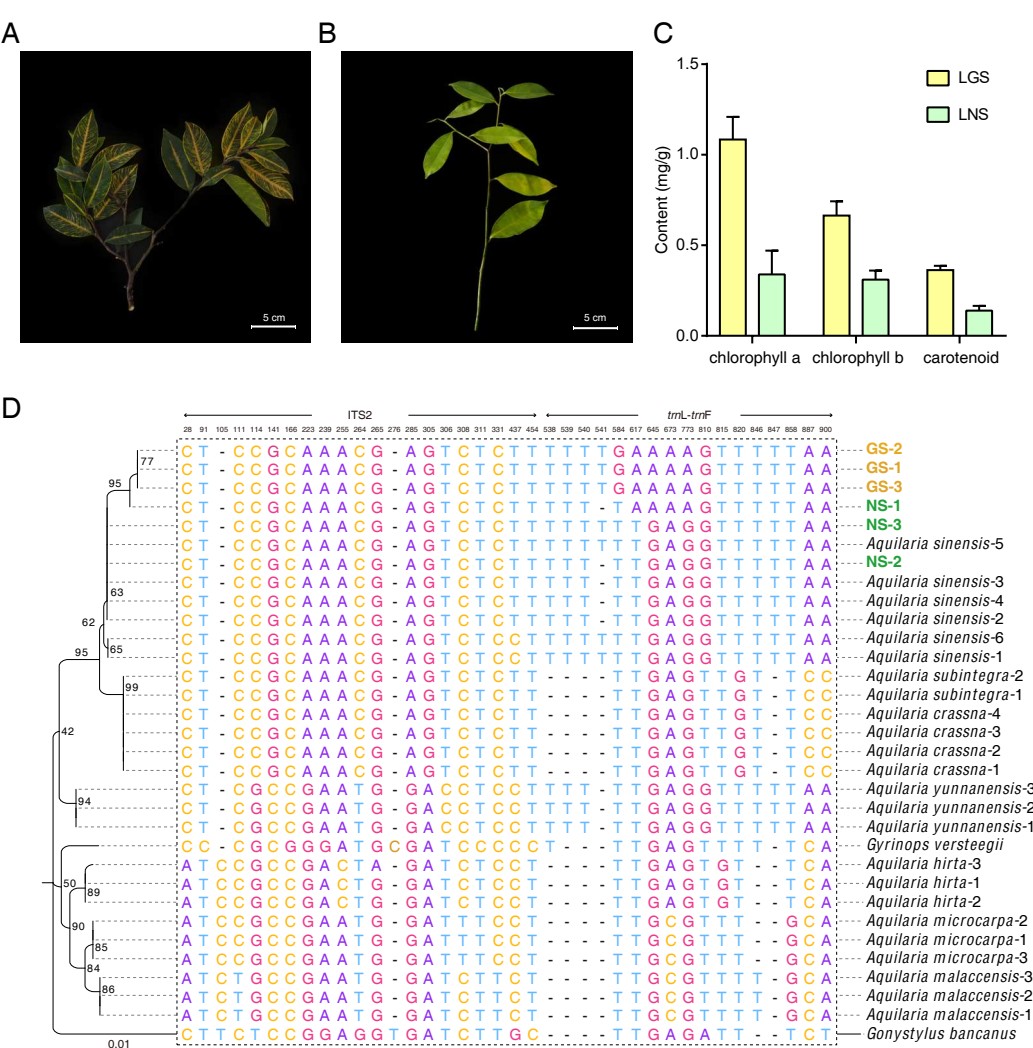

**Figure 1 Plant materials, content of chlorophyll, and the phylogenetic tree for origin identification.**
(A) Golden-vein-leaf sample (GS); (B) normal-vein-leaf sample (NS); (C) the content of chlorophyll of the leaves of GS and NS (LGS and LNS); (D) phylogenetic tree of GS, NS, 8 closely related species, and one distantly related species *Gonystylus bancanus* based on DNA barcode ITS2+*trn* L-*trn* F. The gene accessions of sequences for each species used to construct the phylogenetic tree were listed in Table S1. The scale bar represents 0.01 nucleotide substitutions per site and the numbers next to the nodes are percentages of confidence from 1,000 replicates. The numbers above the different nucleotides indicate aligned positions. Nucleotides specifically exist in outgroup species are not listed.

clades respectively indicating success of the phylogenetic tree construction. The high-level homology of GS and NS sequences with *A. sinensis* verified that they were all *A. sinensis*, but GS samples could not be discriminated clearly with NS samples, which suggested that the variety was not the cause of the GS formation.

## Anatomical comparison of leaves and stems

The photomicrographs of the LGS and LNS were acquired after Safranin O/Fast green staining, of which the anatomical parameters were measured (Fig. 2; Table S2). In the

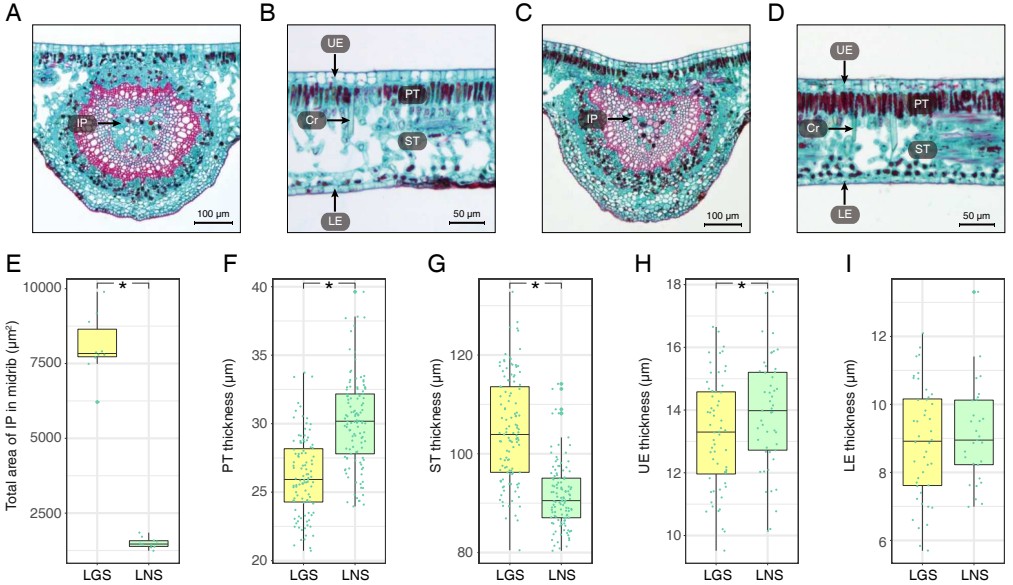

**Figure 2** **Anatomical structures and statistics of main measured parameters of LGS and LNS.** (A) Transverse section of the midrib of LGS (left) and (B) zoomed parts of transverse section of LGS (right); (C) Transverse section of the midrib of LNS and (D) zoomed part of transverse section of LNS; (E) Boxplot of total area of IP in midrib; (F) Boxplot of PT thickness; (G) Boxplot of ST thickness; (H) Boxplot of LE thickness; (I) Boxplot of UE thickness. Two asterisks (**) above two boxes indicates that the P-value is less than 0.0001 and a single asterisk (*) suggests that the P-value is less than 0.05. Those without asterisks mean no statistical significance. (IP, included phloem; UE, upper epidermis cell; LE, lower epidermis cell; Cr, crystal; PT, palisade tissue; ST, spongy tissue).

transverse section of the midrib, the bicollateral bundles were apparent (Figs. 2A and 2C). It was visible that the total area of included phloem in LGS midrib surpassed which in LNS midrib for more than 5 times (Fig. 2E). In addition, the palisade tissue was thicker in LNS whereas the spongy tissue was thicker in LGS (Figs. 2B, 2D, 2F and 2G). Furthermore, the upper epidermis cell thickness in LGS is significantly smaller than LNS, while the difference of lower epidermis cell between LGS and LNS was not significant (Figs. 2H and 2I; Table S2).

Using transmission electron microscopy, the ultrastructure of chloroplast was observed (Fig. 3). It was obvious that the structure of chloroplast in the midrib of LNS was more complete than that in LGS in which the thylakoids were seriously broken (Fig. 3B). Apart from the thylakoid, the starch could be observed in the midrib of LNS (Fig. 3A), however, it was seldom in the midrib of LGS. In addition, we found some particles which were suspected as virus (Figs. 3C and 3D), and some vesicles (Fig. 3C).

With respect to the stems of GS and NS (SGS and SNS), the differences were also significant and the most obvious characteristics observed in transverse surface had been measured and digitized (Fig. 4; Table S3). One of the most obvious features in SGS was that the included phloem tended to connect with each other and their shapes were more irregular than that of SNS (Figs. 4A and 4E) which was typical island-shaped (*Liu et al., 2018*). In addition, the area, length and width of included phloem of SNS were all greater

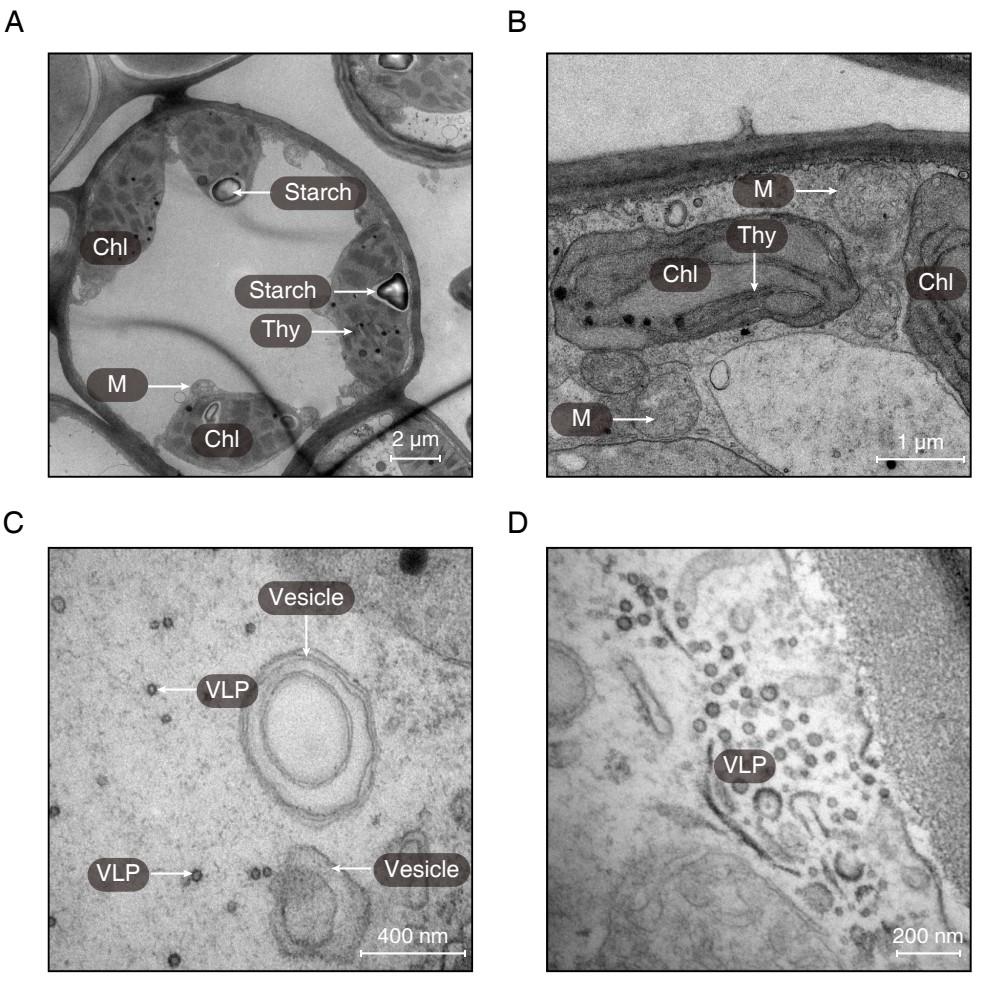

**Figure 3** **The ultrastructures of the midribs of LGS and LNS.** (A) The ultrastructure of the midrib of LNS; (B–D) The ultrastructure of the midrib of LGS. (Chl, chloroplast; M, mitochondria; Thy, thylakoid; VLP, virus-like particle).

than that of SGS (Fig. 4I; Table S3). Because most of the included phloem distributed in xylem, we calculated the percentage of included phloem in xylem of SGS and SNS and found that the number of the former was a little bigger though it was less than 2% (Fig. 4J), indicating that the difference of distribution level of included phloem in SGS and SNS was not as significant as the area. In addition, many crystals could be observed clearly in the pith of SNS (Fig. 4G) while much less in the pith of SGS in which there were more colored grease-like granular substances (Fig. 4C).

The parameters about vessel of SGS were all smaller than SNS (Figs. 4K and 4L; Table S3). In consideration of the features that single vessel area of SNS was larger than that of SGS and the difference between the percentage of vessel to xylem of SGS and SNS was less than 2%, the density of SGS vessel could be higher (Figs. 4B and 4F). Furthermore, the single fiber cell of SGS was significantly smaller than that of SNS (Fig. 4M) and it could be observed intuitively (Figs. 4D and 4H).

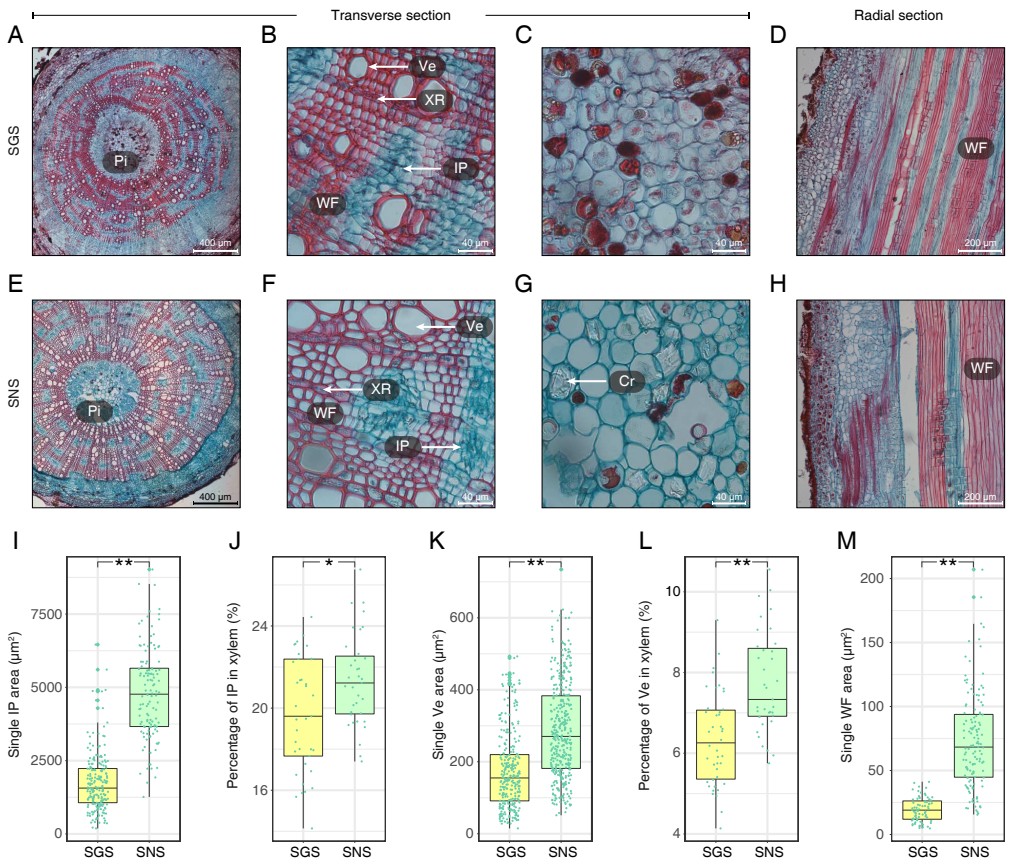

**Figure 4  Anatomical structure and statistics of main measured parameters of SGS and SNS.** (A) Overview of SGS transverse section; (B) a part of transverse section of SGS; (C) transverse section of pith of SGS; (D) radial section of SGS; (E) overview of SNS transverse section; (F) a part of transverse section of SNS; (G) transverse section of pith of SNS; (H) radial section of SNS; (I) boxplot of single IP area; (J) boxplot of percentage of IP in xylem; (K) boxplot of single Ve area; (L) boxplot of percentage of Ve to xylem; (M) boxplot of single WF area. Two asterisks (**) above two boxes indicates that the P-value is less than 0.0001 and a single asterisk (*) suggests that the P-value is less than 0.01 and larger than 0.001. (Pi, pith; IP, included phloem; WF, wood fiber cell; Ve, vessel; XR, xylem ray; Cr, crystal).

## Transcriptome sequencing and differential gene expression profile

Totally 437 million high quality reads (65 G bases) were generated for four samples, LGS, LNS, SGS, and SNS (each with three biological duplicates) (Table S4). Due to the lack of gene annotation information, de novo assemble were performed to get 26,381 unigenes. The contig N50 of all transcripts was 2,168 nt, the average transcript length was 1418 nt, and the complete BUSCOs (Benchmarking Universal Single-Copy Orthologs) was more than half of the total (Table S5; Fig. S2). Functional annotation showed that 18,786 genes could be annotated to at least one of Pfam, SwissProt, TrEMBL, KEGG and GO databases (Fig. S3).

Differential gene expression analyses were performed between LGS and LNS, SGS and SNS using negative binomial models. These genes with multiple test corrected *P*-value less than 0.05 and absolute fold change no less than two were defined as DEGs. In summary,

Peerj

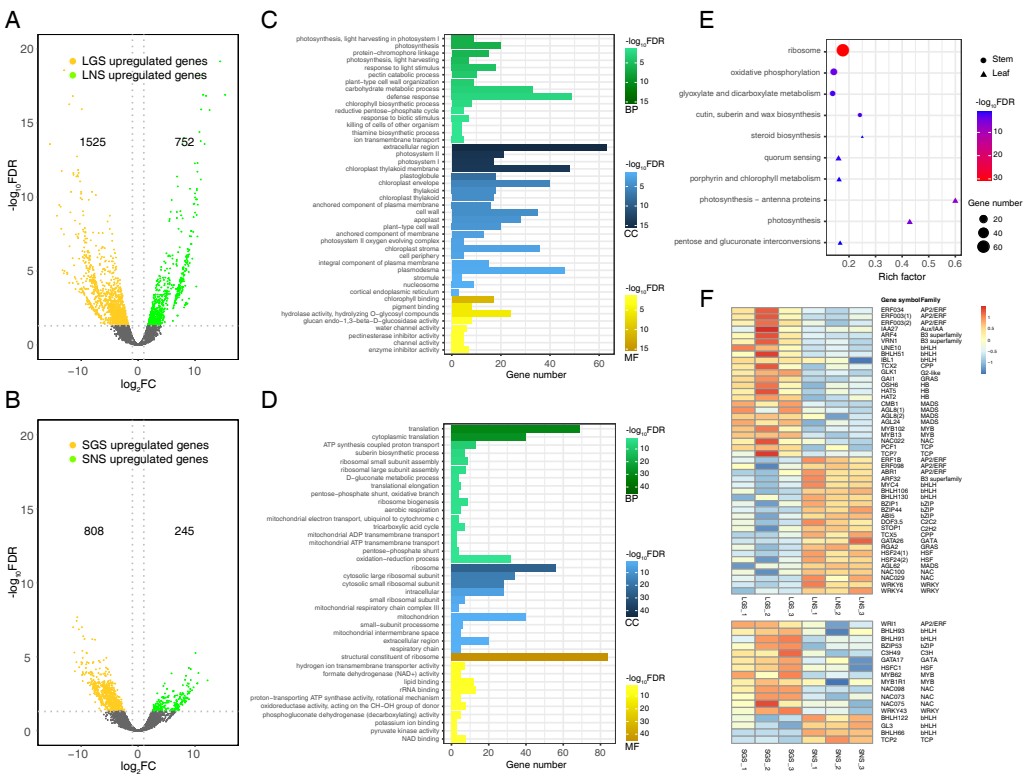

**Figure 5  Distribution and analyses of DEGs.** (A) Volcano plot showing the distribution of DEGs between LGS and LNS; (B) Volcano plot showing the distribution of DEGs between SGS and SNS. Both in (A) and (B), the yellow and green dots in volcano plots indicate the DEGs with log2FC less than -1 or larger than 1 and FDR (false discovery rate) less than 0.05. The numbers above the volcano plots represent the number of upregulated genes for each group. (C) Gene Ontology enrichment analysis for DEGs upregulated in LGS compared to LNS; (D) Gene Ontology enrichment analysis for DEGs upregulated in SGS compared to SNS. The green, blue, yellow gradient color bars indicate the enrichment significance for biological process, cellular component, molecular function respectively in (C) and (D). The length of each bar represents the number of DE genes in each GO term. The color saturation of the bars represents the enrichment significance for each GO term. (E) KEGG pathway enrichment analysis for DEGs upregulated in LGS compared to LNS and the genes upregulated in SGS compared to SNS. (F) Distribution of differentially expressed transcription factors. The color of each cell is on behalf of the Z-score indicating the relative gene expression.

1525 genes were upregulated while 752 genes were downregulated in LGS compared to LNS (Fig. 5A), and 808 genes were upregulated while 245 genes were downregulated in SGS compared to SNS (Fig. 5B). In addition, the samples from same superset could be clustered together using principal component analysis (PCA) of DEGs which were more varianced compared to the no-differential expressed genes (Fig. S4).

## Photosynthesis related genes were enriched in color variation of *A. sinensis* by GO term enrichment analysis of DEGs

Genes upregulated in LGS compared to LNS were significantly enriched in photosynthesis related biological processes including photosynthesis, light harvesting in photosystem I, response to light stimulus (Fig. 5C). Besides, defense response and carbohydrate

metabolic process were two other enriched items occupying the highest proportion of DEGs. Those upregulated photosynthesis related terms may explain the deeper green in LGS and carbohydrate metabolic process was another important term participating in photosynthesis (*Atkin et al., 2000*). In addition, some terms related to cell wall were also enriched, such as pectin catabolic process and plant-type cell wall organization. It was worth considering that the phenotype of midrib may in connection with pectin catabolism. As for cellular component, genes upregulated in LGS enriched in components referred to photosynthesis. Except for "extracellular region", the GO term "photosystem I" and "photosystem II" showed the most significance among these items. Meanwhile, "chloroplast thylakoid membrane" and "chloroplast envelope" covered a large amount of genes. GO terms "cell wall" and "plant-type cell wall" were also enriched which coincides with the terms about cell wall in biological process. For molecular function, the most significantly enriched terms about upregulated genes in LGS were related to chlorophyll such as "chlorophyll binding" and "pigment binding" as well as many genes were enriched into glycosyl hydrolysis relevant terms like "hydrolase activity, hydrolyzing O-glycosyl compounds".

With regard to the downregulated genes (Table S6), the biological process GO terms "response to abscisic acid" and "ethylene biosynthetic process" coincided with phytohormone playing important roles in plant development (*Emenecker & Strader, 2020*; *Iqbal et al., 2017*). Some terms in molecular function involved transporting, such as "xenobiotic transmembrane transporting ATPase activity" and "ATPase activity, coupled to transmembrane movement of substances". In summary, the enrichment terms of downregulated genes were much less than upregulated genes so that there were only three molecular function terms, and two cellular component terms enriched into "plasmodesma" and "plant-type vacuole".

Comparing the GO enrichment analysis for DEGs between SGS and SNS, the significantly enriched GO terms of upregulated genes in SGS were shown in Fig. 5D. For biological process, a lot of upregulated genes in SGS were categorized into the terms which were bound up with translation, energy transporting and metabolism such as "translation", "cytoplasmic translation", "ATP synthesis coupled proton transport" as well as some terms related to ribosome. Focusing on cellular component, many upregulated genes were about ribosome among which the term "ribosome" was the most significant whose gene number was also the most. Moreover, some terms about mitochondrion were enriched too. For molecular function, the most significantly enriched term "structural constituent of ribosome" occupied a dominant quantitative advantage over others echoing with the situation in biological process and cellular component.

The number of downregulated genes which could be enriched was small (Table S7). The terms in cellular component category were not enriched because of the lack of significance and only one term "nitrate transport" was enriched in biological process. The three most significant terms in molecular function were "ADP binding", "sucrose alpha-glucosidase activity" and "double-stranded RNA binding".

## Photosynthesis related pathways were characterized in color variation of *A. sinensis* by KEGG pathway enrichment analysis of DEGs

To analyze the gene functional pathways of DEGs in leaves and stems, the KEGG pathway enrichment analyses were conducted in leaves and stems samples respectively. It was found that only upregulated genes of GS compared to NS were enriched successfully together with the small amount (Fig. 5E). Although the number of genes enriched was much smaller than that of GO enrichment, it also attached importance to analyze the meaningful pathways such as the "photosynthesis - antenna proteins", "photosynthesis", "porphyrin and chlorophyll metabolism" pathways which were related to photosystem obviously. Furthermore, the other pathways associated with saccharide and steroid were worth observing too.

Compared with SNS, 74 genes were enriched in "ribosome" pathway which occupied the largest number in upregulated genes of SGS. The remaining three pathways enriched were "oxidative phosphorylation", "cutin, suberin and wax biosynthesis" and "glyoxylate and dicarboxylate metabolism" respectively.

## Analysis of transcription factors in DEGs

Transcription factors play important roles in plant development in response to biotic or abiotic stimulus through regulating gene expression of abundant genes (*Singh, Foley & Onate-Sanchez, 2002*). To explore the expression of transcription factors, the transcription factors expressed differentially in leaves or stems of samples were screened as shown in Fig. 5F. There were 46 transcription factors expressed differentially in leaf samples and the number in stem samples was 18. In total transcription factors, *bHLH* took up the largest number, and they are important in phytochrome signaling according to the previous studies (*Duek & Fankhauser, 2005*). With respect to other transcription factor family genes, they also played roles in the formation of the phenotype of our samples. For example, the HB family gene *HAT5* (*ATHB1*) could emit its effect in the conversion of palisade to spongy tissue cells in the leaves of *Arabidopsis thaliana* (*Aoyama et al., 1995*). In addition, the G2-like gene *GLK1* was another important transcription factor which could regulate the chloroplast development (*Gang et al., 2019a*) involved in the leaf etiolation (*Xie et al., 2018*). Additionally, since suberin biosynthetic process in SGS could be in connection with the formation of the phenotypes in Fig. 4 because wood fiber tissue accounted for the majority of stem and cutin and suberin play important roles in the formation of wood (*Kolattukudy, 2001*), some *NAC* and *MYB* family genes may be active in the regulation of those process (*Zhong & Ye, 2007*).

## Pathogenesis-related (PR) genes significantly upregulated in golden-vein leaves and stems

The yellow vein symptom of leaves is widely distributed in the plants infected with pathogens. For instance, the genus *Begomovirus* is the largest geminivirus genus containing more than 200 species or members (*Fauquet et al., 2003*; *Fauquet et al., 2008*) of which some viruses have infected a lot of plants with yellow vein symptoms in south China (*Jiao et al., 2013*) which is the place we got our samples from. Dissimilar to the common phenotypes of the infected showing only yellow vein symptom or accompanied with fading, LGS showed

**Table 1  Genes enriched in GO term "response to biotic stimulus".**

| Gene symbol | Description | Log$_2$FC | FDR |
| --- | --- | --- | --- |
| GNS1 | Glucan endo-1,3-beta-glucosidase, basic isoform | −9.8857 | <0.0001 |
| E13B | Glucan endo-1,3-beta-glucosidase | −6.9149 | <0.0001 |
| PRB1 | Basic form of pathogenesis-related protein 1 | −5.2072 | 0.0030 |
| MLP328 | MLP-like protein 328 | −4.9729 | 0.0008 |
| STH-2 | Pathogenesis-related protein STH-2 | −4.4692 | 0.0027 |
| PRR1 | Pathogenesis-related protein R major form | −4.4122 | 0.0286 |

**Notes.**
Negative Log$_2$FC represents the upregulated genes in LGS while the positive value represents the downregulated genes. High absolute value of Log$_2$FC represents a high fold change value. FDR < 0.05 suggests the significant statistical difference in DEGs estimation.

a deeper green beside the yellow vein symptom compared to LNS. In consideration of the enrichment of "defense response" and "response to biotic stimulus" especially the latter in LGS, most of the genes in "response to biotic stimulus" belonged to pathogenesis-related (PR) gene family (Table 1) which was associated to infection and defense (*Van Loon, Rep & Pieterse, 2006*). Using the annotation information, 25 PR family genes expressed differentially in leaves or stems were screened. In those genes, 19 genes were upregulated while one gene was downregulated in LGS compared to LNS as well as five upregulated and two downregulated genes in SGS compared to SNS (Fig. 6A; Table S8). Meanwhile, as shown in Fig. 6B, most of the genes distributed in the third quadrant so that it was clear that most of the PR genes were upregulated in LGS and SGS. Using cluster analysis, 7 genes (*PRB1, PR-1, E137, PER3, PER47, STH-2, NLTP*) were grouped under one cluster in which the gene expressions were both higher in LGS and SGS than their corresponding groups (Fig. 6C). The upregulation of those genes in same expression pattern may be in connection with the infection and the formation of the phenotypes in our samples.

## Quantitative real-time PCR verification of transcriptome data

The expression of those genes in different tissues were identified via qRT-PCR (Figs. 6E and 6F). Based on the enrichment data, those 23 selected genes used in qRT-PCR referred to photosynthesis, cell wall constituent, and some about secondary metabolism (Fig. 6D). Our data revealed that most of the gene expression data in qRT-PCR matched well with transcriptome results (Fig. S5), which verified that photosynthesis related pathways were involved in the color variation phenotypes of *A. sinensis*.

## DISCUSSION

A lot of colored substances are natural colorants showing different colors such as chlorophyll in green, curcuminoid in yellow, carotenoid in yellow to orange to red (*Sigurdson, Tang & Giusti, 2017*). In photosynthesis, the sun-light is captured by chromophores the light-absorbing chemical structure (*Mirkovic et al., 2017*). Chlorophyll, produced from chloroplast which is important in plant development and plant defense (*Lu & Yao, 2018*), could determine the color of plant tissues. When the chlorophyll is broken down, the color of plant tissues would change accompanied with the metabolites showing green, red, or

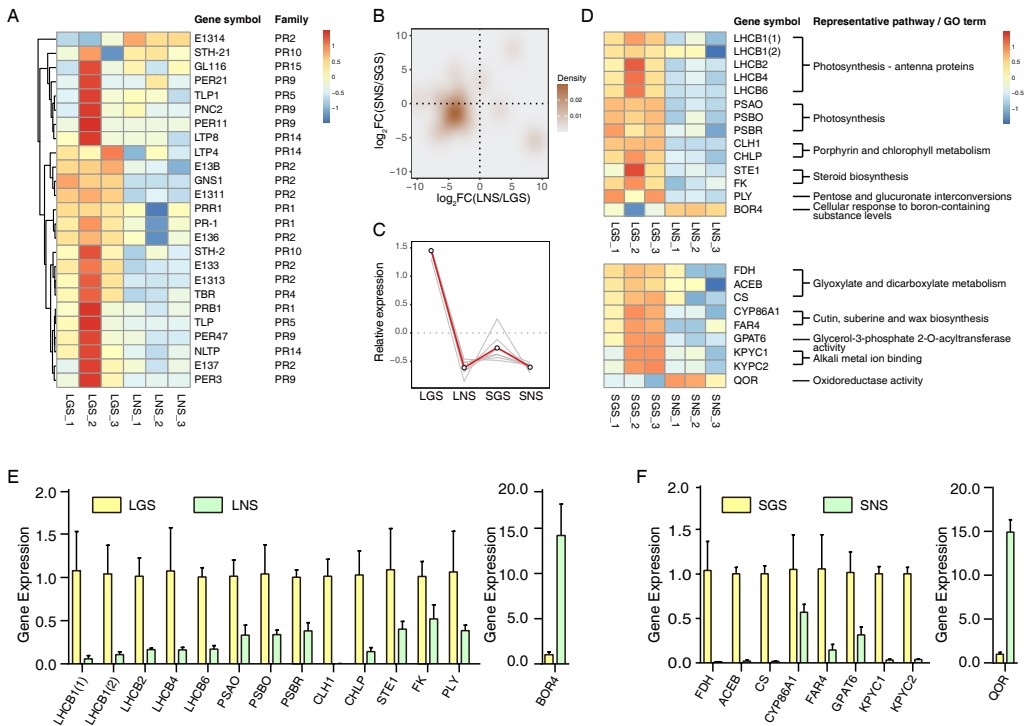

**Figure 6** qRT-PCR verification of selected genes and the heatmaps of selected genes and pathogenesis-related (PR) genes. (A) Heatmap of pathogenesis-related (PR) genes in DEGs. Color saturation represents Z-score scaled gene expression FPKM value. (B) Density map of PR genes. The shade in picture represents the density of gene distribution. (C) Cluster of PR genes with same expression pattern. The gray lines represent the relative expression of each gene and the red line represents the average value for all genes. (D) Transcriptome expression of genes used to be verified with qRT-PCR as well as their annotation and enrichment details. Color saturation represents Z-score scaled gene expression FPKM value. (E–F) Relative expression changes of genes detected using qRT-PCR in leaves (E) and stems (F). In (E) and (F), the genes upregulated and downregulated in the golden samples were divided into two parts. The error bars represent standard deviations (SDs).

other colors even colorless (*Hortensteiner & Krautler, 2011*). Now that golden-vein-leaf sample of *A. sinensis* in this study showing a deeper green is one of the strongest features, which suggested that the genes and pathways related to photosynthesis should be important. As expected, a lot of GO terms and pathways about photosystem were enriched successfully in LGS, among which the upregulated pathways, including "photosynthesis - antenna proteins", "photosynthesis" and "porphyrin and chlorophyll metabolism", should be important to the formation of the dark green leaves in GS (Figs. S6A, S6B and S6C). In the study about birch, the genes related to photosynthesis downregulated in yellow-green leaf whose chlorophyll was less than the normal wild birch, and the parameters including the thicknesses of adaxial epidermis, palisade parenchyma, spongy parenchyma of them were different (*Gang et al., 2019b*). Similar to the condition in the study of birch, our samples also exhibited the anatomical difference in leaves, and it might involve in photosynthesis. Now that transcription factor is a kind of crucial regulator participating in the regulation of gene expression, differentially expressed transcription factors in different leaf sample

groups were screened. Specially, a G2-like gene *GLK1* was upregulated significantly in LGS (Fig. 6F). *GLKs* including *GLK1* and *GLK2* are important transcription factors in leaf growth such as chloroplast development (*Powell et al., 2012*) and leaf senescence (*Rauf et al., 2013*). A research in *Arabidopsis thaliana* has revealed that the induction of *GLK1* could increase the expression of the photosystem and chlorophyll biosynthesis related genes, including *LHCB4, PSBQ, CHLH, CHLM* (*Waters et al., 2009*), all of which also upregulated in our LGS group (Figs. S6A, S6B and S6C). Besides the herbaceous plants, the studies on *GLK* were also conducted in woody plants showing similar results (*Gang et al., 2019a*). In addition, *GLK* genes show positive effects in pathogen resistance in plants (*Han et al., 2016*). A study about *Fusarium graminearum* suggests the *Arabidopsis* with *GLK1* overexpression could increase the expression of some pathogenesis-related genes and be positive in the resistance of fungal pathogen (*Savitch et al., 2007*). According to the properties of *GLK1*, it may be a crucial upstream regulator in the regulation of photosystem and chlorophyll and then leaf-colors.

Chlorophyll breakdown is as important as its synthesis, and the chlorophyll catabolites could be effective in some aspects such as antioxidants (*Muller et al., 2007*) or internal signals sources (*Mur et al., 2010*). Genes upregulated in LGS enriched in the chlorophyll metabolism related pathway and in which the gene like *CLH1* (chlorophyllase) is also active in other chlorophyll breakdown pathway (*Hortensteiner & Krautler, 2011*). Another obvious features in LGS samples are yellow veins. It was a common feature in plants infected with pathogens especially virus like the yellow vein symptoms in *Oxalis debilis* infected with *begomovirus* (*Herrera, Aboughanem-Sabanadzovic & Valverde, 2015*). Regarding the pictures from transmission electron microscope, we observed some particles (Figs. 3C and 3D) suspected as virus because of their shape and the structure of vesicles which is common in the structures beside the virus (*Grangeon et al., 2012; Grangeon et al., 2013; Lerich et al., 2011*). In consideration of the GO enrichment results showing the hints of biotic stimulus and most of the PR genes which are closely related to pathogens defense (*Van Loon, Rep & Pieterse, 2006*) upregulated in LGS, as well as the particle suspected as virus, it is assumed that external factors causing the series of changes could be some kind of pathogens which may be virus.

Stem is the fundamental source part of *A. sinensis* to acquire agarwood resin the precious product (*Liu et al., 2018*). As shown in Fig. 4, the most obvious features in the transverse surface of SGS compared to SNS were the denser wood fiber cell tissue and the thicker cell wall. According to previous studies, cell wall plays an important role in disease resistance (*Underwood, 2012; Molina et al., 2021*), which may explain the phenotype of SGS. In the outer layer, cell wall is covered with pectin and cuticle, and there are suberized wall, plasma membrane, cytoplasm, vacuole from outside of cell wall to inside respectively (*Kolattukudy, 2001*). According to previous researches, some genes from *NAC* or *MYB* families, which were upregulated in SGS (Fig. 5F), have shown their abilities in the regulation of plant cell wall development (*Zhong & Ye, 2007*). Genes upregulated in SGS were mainly enriched in suberin biosynthetic process, and cutin and wax biosynthesis pathways, which suggested that the formation of suberin might change in SGS (Fig. S6D). Taking up half of the above upregulated genes, *CYP86* family genes exhibit important functions in cutin and

suberin and biosynthesis (*Pinot & Beisson, 2011*) in which suberin plays a role in response to physical, chemical or biological stresses as barriers (*Kolattukudy, 2001*). In consideration of the transcription factors screened, *NAC073* (*SND2*) and *NAC075* are related to the regulation of cell wall according to the previous studies (*Sakamoto & Mitsuda, 2015*; *Zhong et al., 2008*). The phenotype of the *SND2* overexpression in *A. thaliana* showed that *SND2* could increase the secondary wall thickening in fiber cells (*Zhong et al., 2008*) which was similar to the photomicrograph of SGS (Figs. 4B and 4F). In addition, the components of the cell walls would even change under the regulation of these *NAC* genes including *SND2* and *NAC075* (*Sakamoto & Mitsuda, 2015*). Therefore, the irregular shapes of included phloem may also be created by the change of components of cell wall leading to the variation of its supporting ability. However, another research claimed that the secondary cell wall thickness of *Arabidopsis* would reduce with the overexpression of *SND2* (*Hussey et al., 2011*). Thus, the roles of transcription factors are still complex to research.

The development of most plants infected with pathogens showing yellow vein symptoms would be influenced and sometimes the plants would die finally. There have been many studies reported that the plants infected with yellow vein related pathogens would grow accompanied with a series of changes like the damage to photosystem in leaves (*Palanisamy, Michael & Krishnaswamy, 2009*), necrotic lesions and veinal necrosis (*Ravelo et al., 2007*). However, if the infected plants can still grow, they could become new sources like ornamental plants (*Valverde, Sabanadzovic & Hammond, 2012*). According to our observation, the trees would not die but grow slower than the normals, so that they could be a new resource applied in breeding. For the moment, the grafting of the trees with unique phenotype is being carried out.

In consideration of the defense response related genes and the pathways about photosynthesis, the pathogen which is responsible for these symptoms may enhance photosystem and promote the metabolism of chlorophyll in tissue specificity. Furthermore, the gene expression variations of stems especially a large number of genes about translation and ribosome suggest that changes in stems are significant and complex. In addition, the grease-like granular substances in the pith in SGS may also hint that the factor causing the symptoms could be potential as a tool to explore the method promoting the formation of agarwood. In the next step, the pathogen, no matter virus, bacterium, fungus or others, should be worthy of isolation and identification, which could be a useful tool to investigate the phenotype formation of *A. sinensis* and expand the potential economic values in the future.

## CONCLUSIONS

Anatomical comparison between golden-vein-leaf sample and normal-vein-leaf sample was conducted in this study, revealing the differences at the microscopic level. In the observation of ultrastructure, the chloroplast was more complete in the midrib of LNS than that of LGS, and some particles suspected as virus were observed. Meanwhile, transcriptome sequencing was performed to analyze the differences of those samples at the molecular level. The genes about photosynthesis and its regulation such as *LHCB4*, *PSBQ*, *CHLH*, *CHLM* and *GLK1*

should be important to the color variation of leaves in *A. sinensis* samples. Analyzing the pathogenesis-related genes in DEGs of all samples, the internal factor leading to the changes might be pathogen infection and seven genes (*PRB1*, *PR-1*, *E137*, *PER3*, *PER47*, *STH-2*, *NLTP*) which were clustered together could be important to it. In summary, our research discussed the internal variation of *A. sinensis* with different phenotypes, and laid the foundation for the studies of the unique phenotype.

### Funding
This work was supported by the Fundamental Research Funds for the Central Public Welfare Research Institutes (ZZ13-YQ-093-C1, ZZ13-YQ-095, ZZXT202112), and Funds for Basic Resources Investigation Research of the Ministry of Science and Technology (2018FY100800). The funders had no role in study design, data collection and analysis, decision to publish, or preparation of the manuscript.

### Grant Disclosures
The following grant information was disclosed by the authors:
Fundamental Research Funds for the Central Public Welfare Research Institutes: ZZ13-YQ-093-C1, ZZ13-YQ-095, ZZXT202112.
Funds for Basic Resources Investigation Research of the Ministry of Science and Technology: 2018FY100800.

### Competing Interests
Ou Huang is employed by Guangdong Shangzhengtang Group Co., Ltd. The remaining authors declare that they have no competing interests.

### Author Contributions
- Jiaqi Gao performed the experiments, analyzed the data, prepared figures and/or tables, authored or reviewed drafts of the paper, and approved the final draft.
- Tong Chen analyzed the data, prepared figures and/or tables, authored or reviewed drafts of the paper, and approved the final draft.
- Chao Jiang and Tielin Wang analyzed the data, authored or reviewed drafts of the paper, and approved the final draft.
- Ou Huang analyzed the data, authored or reviewed drafts of the paper, material resource provided, and approved the final draft.
- Xiang Zhang performed the experiments, authored or reviewed drafts of the paper, material resource collection, and approved the final draft.
- Juan Liu conceived and designed the experiments, performed the experiments, authored or reviewed drafts of the paper, and approved the final draft.

### DNA Deposition
The following information was supplied regarding the deposition of DNA sequences:
DNA barcode sequence genes are available at GenBank: MT548012 to MT548017, and the details of those sequences are provided in Table S10.

## Data Availability

The measured parameters of leaf and stem in Figs. 2 and 4 are available in the Supplemental Files.

The data used in Figs. 5 and 6 is available at NGDC's Genome Sequence Archive: https://bigd.big.ac.cn/gsa/browse/CRA002670.

## Supplemental Information

Supplemental information for this article can be found online at http://dx.doi.org/10.7717/peerj.11586#supplemental-information.

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
