# Peer review of "Comparative anatomical and transcriptomic analyses of the color variation of leaves in Aquilaria sinensis"

_PeerJ, doi:10.7717/peerj.11586_

## Round 0.1 · original submission · Major Revisions

Based on the comments from reviewers, revision is needed for your manuscript prior to acceptance.

Reviewer 1 ·

Basic reporting

1. The authors very well explain the importance of the substance obtained from the endangered tree Aquilaria sinensis. However, very little information is provided on the origin or the region where these two different phenotypes are found and what are the environmental variations - this might give a better understanding of the origin of the new phenotypes and possibly what conditions led to this variation.
2. The authors need some proofreading to correct a few grammatical corrections like in line 46,50,52, 409.

3. Also- in Fig2, the authors mentioned ‘Furthermore, the upper epidermis 193  cell thickness in LGS is significantly smaller than LNS, while the difference of lower epidermis 194  cell between LGS and LNS was not significant ‘ there is some discrepancy in the legend and the figure in fig 2H, 2I (UE vs LE).

4. Could the fig 5A be consolidated anyhow to better represent it? Maybe it can be represented in 2 figures for Stem and leaves separately for all the genes?

Experimental design

1. What kind of tests were performed to note the statistical differences between the groups in fig 2 and fig 3 is not mentioned in the methods.

2. The authors mention that 65Gb odd data was generated from 12 samples that were then assembled using Trinity program to perform a de novo assembly. The average transcript length is shown to be 1418, which very well falls under good transcriptome construction threshold. It might also be worth while to include some statistics of the assembly and the data (e.g concordant/discordant pair/proper pairs, basic config statistics, preprocessing performed -trimmomatic etc)

3. The authors use DESeq for the analysis of differentially expressed genes between the 2 groups. As a QC, if the authors can provide some figure (e.g PCA plot, heatmap) to evaluate how well the samples from same group are clustered together, it will give a good sense of the variability amongst samples based on transcriptomic profiles.

4. In fig 4F and 5B,C - it will also be intuitive to show the expression across the replicates for the 4 groups (LGS,LNS,SGS,SNS) to see the consistency aeross samples within same groups rather than only showing average for each group.

Validity of the findings

1. Although the study dwells on an unexplored aspect of differences in the two phenotypes of an endangered tree due to anatomical and transcriptomic differences, it is not very clear how these findings are relevant to the understand or the differences in survival of the tree or production of agarwood between the two phenotypes

2. The phylogenetic analysis does group the GS and NS with the other Aquilaria sinensis and distinct from Gonystylus bancanus based on ITS2+trnL-trnF. However if we observe the phylogenetic tree the 3 GS are under one branch, however NS2 and NS1 are not. Is this concerning. Is there a possible biological explanation for this?

3.The differences in the anatomy are explained with very high resolution and clear images. But it might be lacking the potential effects of those differences observed. It might be worth discussing how could these differences affect the production of agarwood or survival of the tree

4. The finding of the DEG in LGs vs LNS and SGS vs. SNS, the genes related to photosynthesis is expected due to some compensatory mechanisms to synthesis light owing to their less chlorophyll rich area. However, the pathogenesis related genes up regulation in the SGS and LGS is very interesting. Is there any evidence to the these Golden Vein trees dying earlier than a normal vein tress as suggested by the authors in line 409 ‘the development of most of the plants infected with pathogens showing yellow vein symptoms would be influenced and sometimes the plants would die finally‘

Additional comments

Overall the authors have studied the molecular and the anatomical differences in the normal and golden vein phenotype of Aquilaria sinensis. It is very interesting to study this particular phenotype in a tree that is considered ‘endangered’ and is a producer of a very valued substance ‘agarwood’.

Reviewer 2 ·

Basic reporting

1) Language
(1)The English language is clear and unambiguous. Pay attention to some mistakes of syntax. Lines 115, publishing paper.
2) Background
(1) Your introduction needs more detail about the study of color variegation, especially advances in molecular biology.
3) Article structure, figures, tables. Raw data shared
(1) Figure 2. Where is idioblasts? Please show it in figure legend.
(2) Figure 5. The number order in paper is incorrect.

Experimental design

1) Original primary research within Aims and Scope of the journal.
Yes
2) Research question well defined, relevant & meaningful.
Yes
3) Rigorous investigation performed to a high technical & ethical standard.
Yes, but need more experiments to support the conclusion. The content of chlorophyll and carotenoid? The ultrastructure of chloroplast?
4) Methods described with sufficient detail & information to replicate.
(1) The method of DNA extraction. Line 111, please show the company name of this manufacturer belongs to.
(2) The method of quantitative real-time PCR. Lines 161-162, check the procedure of RT-PCR.

Validity of the findings

1) All underlying data have been provided; they are robust, statistically sound, & controlled
Yse
2) Conclusions are well stated, linked to original research question & limited to supporting results.
Yse

Additional comments

1) Lines 178-180. Please show the result of gel electrophoregram in supplemental figure.
2) Line 220. Each with three biological duplicates or technical duplicates?
3) Lines 238. What is the meaning of DE? The authors need to make sure that they use the full names of all the factors before switching to abbreviations.
4) The species names and gene names should be italic. Please go through the manuscript including references and ensure that all species and gene names are italicized.

---

## Round 0.2 · Minor Revisions

Please make revisions according to the comments from reviewers.

Reviewer 1 ·

Basic reporting

NA

Experimental design

NA

Validity of the findings

NA

Additional comments

All the concerns have been addressed.

Reviewer 3 ·

Basic reporting

1. Figure 2: it would be interesting to specify the P Value that graphs have with a single asterisk.
Not all the parts names of the leaves look good. Change color, size, bold… Especially figure C.
2. Figure 5B: A and B are the same sentence. I understand that in B it should be SGS and SNS. The legend in Figure B also contains the mistake.
3. Figure 6: Better specify the letter E and F. It will help to understand the figure more easily.
4. Table S5: change de novo  de novo
5. Littles mistakes:
a. Line 21 – 22 and 206: different size of letter
b. Line 135: change trnL  trnF
c. Line 196, 219 – 224, 262, 416 – 417: highlight the figures in bold
d. Line 212: change abd  and
6. In the paragraph line 264 “Photosynthesis related genes….” Figure 5C is not referenced.

Experimental design

1. Line 114- 116: the procedure that has been used for sample collection is not well understood. It would be interesting to specify how the collection was, in number of leaves or stems that were collected, mixed ... The experimental design that has been carried out is very important.
2. Line 368: The authors select 22 genes for validation but in graph 6D they are 23. Please revised.
3. Line 371: the authors explain “our data revealed that most of the gene expression data in qRT-PCR matched well with transcriptome results”. The authors could provide some figure o table (e.g correlation graph or similar) to evaluate this.

Validity of the findings

No comments to highlight in this section.

Additional comments

The authors have compared two different types of phenotypes in Aquilaria sinensis at the anatomical and transcriptomic level. It is a very complete and interesting work, in which they provide a large amount of data about these two phenotypes.

Reviewer 4 ·

Basic reporting

The manuscript is written in clear, professional English. The introduction section sets up the premise of the paper very well, and provides enough context about the developments in the field and how this paper serves as an advance. The extensive anatomical comparison of leaves and stems is particularly commendable and shows the comprehensive nature of this study that explores the color variation across two types of leaf samples Aquilaria sinensis

Experimental design

The research question is well defined and meaningful within the scope of the journal.
The investigation performed is rigorous and meets a high level of technical standard.
One minor suggestion is to clearly state which background gene sets are being for GO/KEGG pathway enrichment - it looks like this would be the list of 18,786 genes with functional annotation but should be mentioned.
Also, the in Figure 6E,F the labels for the two y-axes bars should be mentioned, and a short description of what they are should be included in the caption

Validity of the findings

Conclusions are stated clearly and are supported by the results reported by the authors

---

## Round 0.3 · accepted · Accept

Based on recommendations from the reviewers, your manuscript can be accepted for publication.

Reviewer 3 ·

Basic reporting

Thanks for your answer.
No comments to highlight in this section.

Experimental design

Figure S5: it would be interesting to better explain the bar graphs for a better understanding

Validity of the findings

No comments to highlight in this section.

Additional comments

Thanks for anwers.

Reviewer 4 ·

Basic reporting

N/A

Experimental design

N/A

Validity of the findings

N/A

Additional comments

N/A